# Psychological and Psychiatric Comorbidities in Youth with Serious Physical Illness

**DOI:** 10.3390/children9071051

**Published:** 2022-07-14

**Authors:** Sohail Nibras, Rachel Kentor, Yasir Masood, Karin Price, Nicole M. Schneider, Rachel B. Tenenbaum, Chadi Calarge

**Affiliations:** 1Menninger Department of Psychiatry and Behavioral Sciences and Department of Pediatrics, Baylor College of Medicine, 1 Moursund St, Houston, TX 77030, USA; chadi.calarge@bcm.edu; 2Department of Pediatrics, (Primary) Texas Children’s Hospital, Psychology Service, Baylor College of Medicine, Houston, TX 77030, USA; rxkentor@texaschildrens.org (R.K.); klprice@texaschildrens.org (K.P.); nmschnei@texaschildrens.org (N.M.S.); rbtenenb@texaschildrens.org (R.B.T.); 3Department of Public Health, Brown School, Washington University in Saint Louis, One Brookings Drive, St. Louis, MO 63130, USA; yasirm@wustl.edu

**Keywords:** anxiety, chronic illness, delirium, depression, medically ill, palliative care, psychological distress

## Abstract

An estimated one in six children in the United States suffers from a mental disorder, including mood, anxiety, or behavioral disorders. This rate is even higher in children with chronic medical illness. This manuscript provides a concise review of the symptoms that comprise mental conditions often observed in children with chronic illness or at the end of life. It further provides some guidance to help clinicians distinguish normative from pathological presentations. Evidence-based psychotherapy interventions, potentially applicable to the acute inpatient setting, are briefly summarized. Broad recommendations are made regarding both psychotherapeutic as well as pharmacotherapeutic interventions, with a review of common or serious medication side effects. Finally, delirium recognition and management are summarized.

## 1. Psychological Distress among Children with Serious and/or Terminal Illness

One in six children (ages 6 to 12 years old) and adolescents (ages 13 to 17 years old, henceforth both referred to as children unless the data are specific to one age group) in the United States suffers from a mood, anxiety, behavioral, or other psychiatric disorder [1]. Globally, subclinical distress is even more prevalent. Unsurprisingly, children with chronic medical conditions, including cancer, sickle cell disease, cystic fibrosis, and pain, report psychological distress at a higher rate than those without [2,3]. Similarly, anxiety disorders and disruptive behavior, including defiance, oppositionality, argumentativeness, and emotion dysregulation, are also common among children with chronic illness, with prevalence rates ranging from 7% to 40% [4,5]. Notably, these conditions may be premorbid and/or inadvertently reinforced by well-intentioned but ultimately unhelpful parenting behaviors. Specifically, parents of children with a chronic illness often have difficulty with limit setting and engage in permissive practices to offset the children’s adversities, potentially leading to increased noncompliance and “acting out” behaviors (e.g., arguing, defiance, and being disruptive) [6].

Given the stressful nature of hospital admissions, children newly diagnosed with a health condition may also suffer from an adjustment disorder, comprising emotional and/or behavioral responses beyond what is expected given a specific stressor [7,8]. Moreover, many life-saving medical procedures can be experienced as traumatic by children, with up to 62% of critically ill children in the pediatric intensive care unit (PICU) endorsing symptoms of post-traumatic stress [9]. In fact, post-traumatic stress disorder (PTSD) is more prevalent among youth with cancer compared to youth facing other non-medically related traumatic events [10].

A certain level of distress in a hospital setting is both appropriate and expected. As such, distinguishing between distress that is normative compared to one warranting intervention is a common challenge for clinicians working with chronically and/or critically ill children. Further complicating assessment is the consideration of how the underlying medical condition and its treatment may impact cognitive and emotional functioning. For example, many medical conditions produce somatic sensations that are identical to the physical manifestation of anxiety symptoms, making it difficult to ascertain what is directly related to the illness as opposed to being due to anxiety. As a result, multidisciplinary assessment is critical for accurate and comprehensive evaluation of a patient’s distress [4].

## 2. Importance of Addressing Mental and Behavioral Health

The relationship between psychiatric and other medical conditions is bidirectional, with this comorbidity being associated with greater functional impairment than would be expected with either alone [11]. Remediating comorbid psychiatric conditions can improve engagement with medical care, treatment adherence, symptom management, and overall quality of life [12,13,14,15]. Moreover, resiliency can be bolstered by skills-based intervention to promote tolerating uncertainty, increasing flexibility, and managing pain and other burdensome physical symptoms.

## 3. Anxiety

Transient worries and fears are common in children, particularly those experiencing stressful life events. However, anxiety is considered problematic when symptoms are excessive, inappropriate, or persistent, and associated with significant impairment or distress. Chronic medical conditions substantially increase the risk of experiencing anxiety symptoms in children, with an estimated 20% to 50% of those with asthma, type 1 diabetes, epilepsy, inflammatory bowel disease, juvenile idiopathic arthritis, congenital heart disease, and sickle cell disease developing an anxiety disorder [11,16]. Anxiety disorders most commonly seen in children with medical illnesses are listed in Table 1.

### 3.1. Psychotherapeutic Interventions for Pediatric Anxiety

For children able to tolerate procedures but who experience anticipatory anxiety, coping strategies, rather than exposure-based interventions, are warranted. This may include relaxation strategies (e.g., guided imagery, diaphragmatic breathing), distraction, etc. For a clinically significant anxiety disorder, however, targeted psychological interventions are indicated.


*Case vignette: Charlie is an 8 year old with rhabdomyosarcoma admitted for management of acute chronic pain following surgical resection of his tumor. Consistent with separation anxiety, Charlie expresses fear that he will experience an acute medical event when his mother is out of the room, or that she will get lost in the hospital. He requires his mother to be in bed with him to fall asleep and begs, cries, and eventually has a tantrum if his mother attempts to leave his room, even when other adults are present. In order to decrease upset feelings and avoid behavioral outbursts, Charlie’s mother tries to “sneak out” of Charlie’s room when he is asleep. She also lays in the bed with him all night long, sacrificing her own sleep.*


#### 3.1.1. Cognitive Behavioral Therapy (CBT)

Amongst different psychotherapies, CBT has the most robust evidence for childhood anxiety disorders and has been adapted to the medical setting [17]. CBT interventions are based on an understanding of the interaction between anxiety and avoidance. For example, caregiver behavior may reduce child’s anxiety in the immediate term, but unwittingly maintain or reinforce it in the long term.

It is “natural” for Charlie to seek to reduce the anxiety/distress related to anticipated or actual separation (i.e., *avoidance*) by insisting on his mother’s presence or co-sleeping with him and for the mother to accommodate him. While avoidance and accommodation will result in symptom relief acutely, they will invariably perpetuate Charlie’s anxiety in the long run. CBT comprises a set of techniques involving psychoeducation about anxiety and the factors that maintain it (including accommodating behaviors), gradual or stepwise exposure to anxiety-provoking situations, and recognizing and modifying maladaptive thought patterns (cognitions) that accompany anxiety [18].

In the case of Charlie, the psychologist may collaborate with Charlie and his mother to target a goal of Charlie falling asleep without his mother in his hospital room. A “fear hierarchy” or “bravery ladder” would be developed, detailing small “bravery practices” that can be implemented in a graduated manner. Exposure practices start with easier tasks (e.g., mother sitting in a chair holding Charlie’s hand while he falls asleep) and progress to more anxiety-provoking situations (e.g., mother leaving the room for a few minutes while Charlie lays in his bed). Rewards are often used to increase motivation to engage in exposure practices, with younger children benefitting from immediate reinforcement (e.g., stickers) and older children typically using a token economy (e.g., points to trade in for later rewards). Through this process, Charlie gradually habituates to sleeping alone as his fears are disconfirmed. Naturally, for exposure to be effective, a strong alliance with the caregiver is critical. Their own distress about seeing their child suffer must be addressed by engaging them to help their child “be brave and defeat anxiety”.

Similarly, brief cognitive interventions can assist with excessive worry. At a minimum, children and families are coached in recognizing “worry thoughts” when they occur and labeling them as such, in a non-judgmental manner (That sounds similar to a worry thought). They then learn that worry thoughts are most often “tricks” that anxiety plays on them to make them feel as if they are not safe. They can turn their attention to something more important or interesting. Of note, CBT does not typically use rationalization or logic as a defense against worry thoughts (e.g., the mother has been with the child in the hospital for the past 2 months and has never gotten lost once) because anxiety is generally not rational. In addition, children and parents are taught that pushing worry thoughts away or trying hard not to have worry thoughts most often has the paradoxical effect of increasing the frequency of the thought. Here, again, caregivers are educated about the unintentional reinforcing effect of accommodations and coached to use more adaptive approaches. For instance, when a child asks a question repeatedly due to anxiety, the parents should answer factually one time. Any question asked more than once is labeled as reassurance seeking, and the caregiver either tells the child she will not answer, or just ignores it and redirects the child’s attention.

#### 3.1.2. Acceptance- and Mindfulness-Based Psychotherapy

Growing evidence has shown that acceptance- and mindfulness-based therapeutic approaches, including Acceptance and Commitment Therapy (ACT) and Mindfulness-Based Stress Reduction (MBSR), effectively treat children with a variety of psychological and physical presentations [19,20,21]. The goal of ACT is to cultivate values-based living, regardless of unwanted internal thoughts, emotions, or bodily symptoms [22]. Rather than seeking to evaluate and alter distorted thoughts and internal experiences, as is the focus in CBT, ACT mindfulness-based interventions aim to promote psychological flexibility. Other components of ACT include acceptance, cognitive “defusion”, being present, self-as-context, values, and committed action [22]. Specifically, increasing psychological flexibility helps children increase their ability to tolerate the loss of control and unpredictability often associated with unexpected medical procedures and hospital stays.

In the case of Charlie, he would be taught how our “funny minds” can come up with unpleasant thoughts, memories, and images that are no more or less true than pleasant ones. Behavioral exposure would be incorporated into the treatment plan in much the same way as would be carried out in CBT. However, the focus during exposure would be on noticing the internal experiences (including thoughts and emotions as well as bodily sensations) in a dispassionate way. One example exercise is “The Thought Parade” [20], wherein Charlie would focus his attention on his breath and then imagine watching his thoughts go by similar to floats in a parade. He would be encouraged to notice that some “floats” are bigger and brighter, some move slowly or even get stalled. If he finds himself swept up in the parade, he would imagine himself returning to the sidewalk to resume watching.

Acceptance- and mindfulness-based interventions are particularly well-suited to children with chronic medical conditions. In chronic pain, for example, youth often withdraw from people and activities that are meaningful to them, due to pain itself and/or fear of pain. A clinician utilizing ACT would help the child evaluate the workability and costs of their pain avoidance. Rather than focusing on controlling the pain at any cost, the goal would be to live a meaningful life even with pain (e.g., a child going to school even though walking in the hallways may temporarily increase pain, because seeing friends and keeping up academically is important to them).

### 3.2. Pharmacological Interventions for Pediatric Anxiety

For moderately severe to severe anxiety disorders, psychotropic medications may prove necessary. In fact, at times, even for isolated anxiety symptoms (e.g., insomnia), the use of medications may also be indicated. Selective serotonin reuptake inhibitors (SSRIs) are recommended as first-line pharmacotherapy for anxiety disorders in youth, followed by serotonin and norepinephrine reuptake inhibitors (SNRIs) [23]. As a general rule of thumb, medications should be started at the lowest possible dose and increased based on efficacy and tolerability. While about 50% of the ultimate response is observed within two weeks of treatment initiation, maximum clinical effectiveness of antidepressant medications may take up to 6 to 8 weeks [24]. Moreover, due to possible emergent suicidality, patients should be monitored closely [23]. The treatment course should last for 8 to 12 months after symptom remission, with medication discontinuation ideally reserved to a time of low stress and carried out over an extended period, while monitoring for relapse or emergence of discontinuation symptoms [23].

Because SSRIs are thought to have comparable efficacy in anxiety disorders, the choice of specific agents is based on its pharmacokinetic properties and side effect profile. Importantly, SSRIs vary in their potential to inhibit the P-450 cytochrome enzymes, including 3A4, implicated in the metabolism of many medications, including macrolide antibiotics and azole antifungal agents [4]. As such, the potential for drug–drug interactions and QTc interval prolongation should be kept in mind. Citalopram and escitalopram cause the least drug–drug interactions while sertraline is the least likely to prolong QTc interval [4]. In contrast, duloxetine should be considered when there is comorbid neuropathic pain or activation from SSRIs [24].

Benzodiazepines are often used for acute anxiety states or panic attacks, including those triggered by medical procedures, given their relatively rapid onset of action [23]. For more extended symptom control, longer-acting agents, such as clonazepam, should be considered [4]. Lorazepam, oxazepam, and temazepam are recommended for use in patients with liver failure as they are metabolized by conjugation and excreted renally. In children undergoing chemotherapy, lorazepam may have the added benefit of reducing nausea and vomiting [25]. Of note, the extended use of benzodiazepines is associated with tolerance and a significant risk of physical and psychological dependence [26]. Clinicians should be mindful of benzodiazepine withdrawal particularly when transitioning patients from the PICU to lower acuity units [4]. They should also avoid benzodiazepines in patients at a high risk of developing delirium.

Additional options to treat anxiety disorders include α-2 agonists, antihistamines, buspirone, tricyclic antidepressant, and atypical antipsychotics, which are listed in Table 2. However, at that point, consulting a child and adolescent psychiatrist may be advisable.

## 4. Depression

Major depressive disorder (MDD) is the fourth most common cause of disability and impairment in youth aged 15 to 19 years, and the fifteenth in those aged 10 to 14 years [27]. Depressive episodes in children are associated with significant morbidity and mortality, with suicide being the second leading cause of death in this 10 to 19 age group in the United States. Depression has a multifaceted and diverse etiology resulting from interactions between biological, psychological, and environmental factors [28].

In medically ill children, various biological factors (e.g., inflammation or disruption to metabolic pathways impacting neurotransmitter signaling) may contribute to the onset of MDD [29]. In addition, youth with chronic illnesses face a host of barriers and adversities, including social isolation, inability to engage in normal activities, parental attitudes, and behaviors (e.g., overprotection), etc. [30,31]. These factors work in consort with biological factors to precipitate and perpetuate depressive disorders [32]. Naturally, the child’s developmental stage influences his or her experience of their medical illness and the expression of depression. As such, it is imperative not to confound depressive symptoms in an emotionally healthy youth who is having reactions to a medical illness with the onset of MDD. On the other hand, it is equally important to identify when MDD is present given its potential to impact quality of life and cause treatment nonadherence, exacerbating the medical illness, healthcare use, and school absences (Table 3) [33,34].

### 4.1. Psychotherapeutic Interventions for Pediatric Depression

A variety of evidence-based psychotherapeutic treatments for depression are available (American Psychiatric Association, 2000), chief among them being CBT, shown to help in several chronic conditions, including chronic pain, diabetes, irritable bowel syndrome, and polycystic ovarian syndrome [37,38]. Similarly, mindfulness-based therapies (i.e., ACT, MBSR, and Dialectical Behavioral Therapy or DBT) have demonstrated efficacy in treating depression among adolescents with chronic illness [39,40,41].

Nonetheless, optimal approaches can be difficult to implement in the context of serious medical illness, given the number of barriers such as compromised immunity, increased fatigue and nausea, potential physical restrictions, required absences, and impaired cognitive processing. As such, flexibility in care delivery is paramount and may include delivering the intervention via telehealth, shown to be an effective mean to conduct therapy [42]. Of course, the COVID-19 pandemic has made the use of telehealth services commonplace.

### 4.2. Psychopharmacological Interventions for Pediatric Depression

No randomized controlled studies have examined the efficacy of psychotropics in medically ill youth with MDD. Recommendations are therefore based on findings in medically healthy children, where SSRIs are considered first line. Fluoxetine (ages 8–17 years) and escitalopram (ages 12–17 years) are FDA approved for the treatment of depression in youth. Again, clinicians should consider drug interactions when choosing an antidepressant for the medically ill child (Table 2 and Table 4).

## 5. Psychological Considerations at the End of Life

Youth with terminal diagnoses face cumulative losses throughout their illness course, from the loss of control over their own body, to the loss of personal identity and roles, loss of future goals, and ultimately the anticipated loss of relationships upon death [46]. These losses can manifest in *existential distress*, a concept frequently addressed in the palliative care literature [47,48]. Dying children grapple with questions of “why me?” and affronts to their world views. They may question the meaning of life and have difficulty reconciling with their faith (e.g., “why would God do this to me?”). As their personal agency and control diminish with disease progression, they may increasingly struggle with feelings of isolation and concerns about their family’s well-being and fear of being a burden. Medical teams can mitigate some of this distress by engaging youth in discussions about their achievements in life as well as those they are yet to accomplish, fostering hope (if not for cure, for comfort and meaning), learning about their personal relationships, and more [47].

It is common for children to experience increasing anxiety as their disease progresses and their physical condition declines [48]. This is due in part to the physical experience of symptoms as death approaches. Increased sleepiness, changes in respiration, and decreased appetite are salient markers to patients of their own decline [49]. Importantly, depression, anxiety, and fear are more prevalent among adolescents than in younger children with a terminal diagnosis, likely reflecting their capacity for more abstract thinking [50]. That said, while some level of distress is a normative and appropriate response to death, suffering, and the unknown, assuming that high anxiety and depression are inevitable is potentially harmful to dying patients as it may decrease team efforts to address these symptoms [48]. Mitigating avoidable distress at the end of life for youth is as crucial as minimizing physical pain to achieve a “good death” [51].

Given the nuances of psychological distress at the end of life, psychotherapeutic intervention must be tailored accordingly. This sometimes involves wholly reversing interventions that are used among youth without a terminal diagnosis. In the case of Charlie, highlighted above, the intervention sought to reduce parental accommodation of Charlie’s anxious behaviors (i.e., encouraging his mother to leave the room so he can spend greater periods of time alone). For a dying child, it would instead be appropriate to foster such continued presence and reassurance of closeness. The emphasis changes to symptom management rather than treating the underlying cause of the symptoms. Psychoeducation to patients and families can be an important intervention target, such as helping children recognize the reciprocal relationship between anxiety and dyspnea. Psychoeducation and anticipatory guidance can be coupled with relaxation strategies, augmented further by a combination of benzodiazepines and opioids. Accordingly, attempts at cognitive restructuring would be inappropriate for patients with advanced disease and/or those approaching the end of life, given that their intrusive thoughts and ruminations are realistic and inevitable [52]. Instead, acceptance- and mindfulness-based approaches are indicated. One example is helping youth focus on the present moment rather than anticipating their future decline (e.g., “I don’t know how much longer I’ll be able to walk, but I can walk today”). Values-based living becomes paramount at the end of life to help youth and their families navigate decisions about their goals of care and identify opportunities for “committed action” to ensure their remaining time is spent in a manner most meaningful to them.

### Special Considerations for Adolescents and Young Adults at the End of Life

Adolescents and young adults (AYA) broadly comprise individuals between the age of 15 and mid-twenties to late-thirties, depending on the disease, country, and/or individual institution. This group faces unique challenges and, as such, warrants tailored support. Serious illness disrupts normative developmental milestones and tasks of adolescence, including identity development, increased independence and autonomy, exploration of sexuality, body image and self-concept, and development of romantic relationships, as well as future planning for career and family goals [53]. Accordingly, AYAs demonstrate higher rates of depression and anxiety at the end of life when compared to younger children, expressing increased feelings of isolation as curative outcomes become less likely [53]. Twenty percent of patients presenting to an AYA-specific palliative care program reported a premorbid history of mental illness, with more than one-third endorsing clinically significant symptoms of anxiety and/or depression at their first clinic visit [54]. In the last month of life, AYAs most commonly endorse feelings of sadness, anxiety, fear of being alone, fear of dying, fear of pain, and guilt [55]. In addition to psychological symptoms, AYAs with advanced cancer also demonstrate a higher prevalence of complex pain and greater opioid usage when compared with older adults [55], further reflecting and contributing to overall symptom burden and total distress.

Psychiatric and psychotherapeutic interventions, as described above, can help AYAs with terminal illness. After pain medication and anti-emetics, anxiolytics are the third most common class of medication prescribed to AYAs with cancer at the end of life, particularly among older adolescents. In fact, more than two-thirds of patients aged between 18 and 21 are taking anxiolytics in the period immediately preceding death [56]. Early research also suggests that integrating psychiatry services within an AYA-directed palliative care program may help to improve and/or prevent the worsening of depressive and anxiety symptoms among patients with advanced cancer in as little as one visit [54].

## 6. Delirium

Delirium is a neurocognitive disorder, characterized by its acute onset, with fluctuating impairment in attention, cognition, arousal, along with sleep dysregulation and possible perceptual disturbances [7]. The majority of youth with delirium exhibit affective lability, anxiety, and irritability [57,58]. Symptoms may be particularly prominent in the evening and at night, a phenomenon called “sundowning”. The clinical presentation may be characterized by withdrawal (i.e., hypoactive delirium), agitation and even aggression (i.e., hyperactive delirium), or a mixture of both (i.e., mixed). Hyperactive delirium most often comes to clinical attention, but the mixed and hypoactive presentations are far more common [7,59,60,61].

Delirium is a multifactorial condition that results from the direct physiological changes associated with medical or surgical conditions, substance use or withdrawal (including medications), nutritional deficiencies, or toxin exposure, among others [62]. Three mechanisms are thought to mediate the onset of delirium, including alteration in neurotransmitter signaling, neuroinflammation, and increased oxidative stress [63]. Even after accounting for illness severity, delirium is associated with increased short- and long-term morbidity, leading to higher healthcare resource utilization and expenditure, and mortality [7,61,64,65,66].

Given that delirium pathophysiology is directly linked to the underlying medical condition, its prevalence is tightly related to the population under study, estimated to be as high as 49% to 57% in children with cyanotic disease, longer durations of mechanical ventilation, and long cardiopulmonary bypass times [67,68]. Risk factors in children include an age of less than 5 years, pre-existing neurodevelopmental delays, acute pediatric disease, presence of a pre-existing pediatric condition, malnutrition states, and mechanical ventilation. Anticholinergic drugs, benzodiazepines, opioids, immobilization, and restraints are among the modifiable risk factors for delirium [61,63,67,69].

The assessment for pediatric delirium should include a comprehensive medical history and review of the medical record, laboratory, and imaging test results, in addition to soliciting parents and nursing staff observations to capture fluctuations in the level of consciousness and attention. Because delirium, particularly the hypoactive type, is underdiagnosed, a validated screening tool should be used, ideally completed more than once daily to capture waxing and waning symptoms. Widely used assessment tools include the Cornell Assessment for Pediatric Delirium (CAPD/CAPD-R), the Preschooler and Pediatric Confusion Assessment Method for the Intensive Care Unit (pCAM-ICU), and the Vanderbilt Assessment for Delirium in Infants and Children [70,71].

### 6.1. Delirium Management

Managing delirium starts with prevention, which includes scheduled assessments and identifying and mitigating risk factors, such as perioperative hypotension and hypoxemia as well as minimizing the use of sedatives/hypnotics. Nonpharmacological and pharmacological approaches may be appropriate (Table 5) [61,72,73].

### 6.2. Pharmacological Interventions for Delirium

There is no FDA-approved medication to treat delirium. Older guidelines have supported the use of antipsychotic medications in the treatment of pediatric delirium [58,61,63,73,74,75,76,77]. However, recent evidence in adults has converged on showing that antipsychotics largely fail to prevent delirium or shorten its duration [78,79]. In fact, more recent guidelines have recommended against the use of antipsychotics in the management of delirium in adults, reserving them to selected cases [80]. Instead, clinicians are encouraged to use the ABCDEF bundle (Table 6) and consider dexmedetomidine, shown to be effective in delirium prevention and treatment in adults [81]. Melatonin is often used to normalize the sleep cycle. When antipsychotics prove necessary, adverse effects should be monitored, including QTc interval prolongation, dysrhythmias, and extrapyramidal signs.

In summary, psychological and behavioral disturbances in children suffering from a severe and debilitating medical condition are common and compound the distress and impairment they are enduring. As such, vigilance about screening for such conditions and prompt treatment are paramount to lessen the burden these children are struggling with. Future research should more specifically examine the efficacy of psychological and pharmacological interventions in children with comorbid medical and psychiatric conditions, who have traditionally been excluded from clinical trials. This would further strengthen the evidence base to inform clinical management.

## Figures and Tables

**Table 1 children-09-01051-t001:** Examples of anxiety-related symptoms exhibited in a medical setting.

DSM-5 Description *	Developmentally Typical/Appropriate	Signs of Anxiety Disorder	Presentation in Medical Setting
Specific phobia: Marked fear or anxiety about a specific object or situation (e.g., blood-injection-injury type)	Young child cries in anticipation of vaccination during routine well child visit	Child anticipates the need for an injection well ahead of a scheduled visit; seeks reassurance from parent; refuses to get in the car; requires parent or staff to restrain her	Child screams, lashes out, and tries to escape necessary blood draws; requires repeated physical restraint
Separation anxiety disorder: Developmentally inappropriate and excessive fear or anxiety concerning separation from those to whom the individual is attached	Child experiences tearfulness and clinginess on the first day of school or when staying with a sitter	Child refuses to attend school or requires escort from the car to the school building; child will not sleep alone; child must be in a room with an adult at all times, even at home	Parent must sleep in the hospital bed with the child; child will not allow parent to leave the room; child requires sedation before being taken from the parent’s presence for procedures
Generalized anxiety disorder (GAD): Excessive anxiety and worry (apprehensive expectation) about different events or activities. The child finds the worry difficult to control, and it is associated with behavioral or physical symptoms	Child has occasional difficulty falling asleep due to worries about grades or tests	Child is described as a “worry wart”, with worries across different domains; with insomnia, fatigue, restlessness, irritability, trouble concentrating, or muscle tension; excessive reassurance that fears will not be realized is needed	Child exhibits excessive worries, particularly treatment failure or death; requires constant reassurance; asks questions repeatedly; has difficulty falling asleep; demonstrates symptoms inconsistent with or in excess to what may be caused by the medical condition or its treatment (e.g., headache, stomachache)
Panic attack: An abrupt surge of intense fear or discomfort, during which time the following may be experienced: accelerated heart rate, sweating, shaking, shortness of breath, chest pain, sense of choking, GI distress, dizziness, feeling hot or having chills	Child experiences physiological sensations in anxiety-provoking situations	Child experiences physiological sensations in response to stress or “out of the blue”, along with catastrophic thoughts about symptoms and avoids situations in which similar symptoms may be anticipated	Child experiences physiological sensations catastrophic thoughts, and avoidance Symptoms may mimic those of medical illness Symptoms due to the underlying illness may trigger full panic attacks

* Adapted from the Diagnostic and Statistical Manual of Mental Disorders—5th Edition (DSM-5).

**Table 2 children-09-01051-t002:** Medications for anxiety and depression in children.

Name	Dose Range	Starting Dose	FDA Approved (Age Range, Years)	Most Pertinent Side Effects
Selective Serotonin Reuptake Inhibitors (SSRIs)				
Citalopram (Celexa)	10 to 40 mg	10 mg	None	Headaches, gastro-intestinal side effects, feeling jittery, disinhibited, activated, irritability, impulsivity, agitation, suicidality.
Escitalopram (Lexapro)	5 to 20 mg	5 mg	Depression (12–17)	
Fluoxetine (Prozac)	10 to 20 mg	10 mg	Depression (8–17) OCD (7–17)	
Fluvoxamine (Luvox)	50 to 200 mg	25 mg	None	
Paroxetine (Paxil)	10 to 40 mg	10 mg	None	
Sertraline (Zoloft)	12.5 to 200 mg	25 mg	OCD (6–17)	
Serotonin norepinephrine reuptake inhibitors (SNRIs)				
Duloxetine (Cymbalta)	30 to 60 mg	30 mg	Anxiety (7–17)	Cardiovascular and hepatic side effects.
Venlafaxine (Effexor XR)	37.5 to 225 mg	37.5 mg	None	
Other Antidepressants				
Mirtazapine (Remeron)	7.5 to 45 mg	7.5 mg	None	Somnolence, agranulocytosis, QTc prolongation, and weight gain.
Tricyclic Antidepressants				
Amitriptyline (Elavil)	10 to 200 mg	10 mg	Depression (12+)	Cardiovascular and anticholinergic side effects. May be lethal in overdose.
Desipramine	25 to 100 mg	25 mg		
Nortriptyline	1 to 3 mg/kg/day		Depression (6+)	
Benzodiazepines				
Lorazepam (Ativan)	0.5 to 2 mg	0.5 mg	None	Sedation, confusion, disinhibition, and/or paradoxical activation particularly in youth with CNS dysfunction.
Clonazepam (Klonopin)	0.5 to 1 mg	0.5 mg	None	
Antihistamines				
Hydroxyzine (Atarax, Vistaril)	50 mg (age <6) 50 to 100 mg (6+)	50 mg	None Approved for adult GAD	Sedation, fatigue, dizziness, anticholinergic side effects, and paradoxical activation (in younger children).
α2 Agonists				
Clonidine	0.05 to 4 mg	0.05 mg	None	Sedation and hypotension. May reduce risk of delirium.
Guanfacine (Tenex)	0.5 to 4 mg	0.5 mg	None	
Atypical Antipsychotics				
Olanzapine, Risperidone, Quetiapine, Aripiprazole	May be used at low doses.			
Antiepileptics				
Gabapentin, Pregabalin	Have been used to treat anxiety disorders in adults.			

**Table 3 children-09-01051-t003:** Differential diagnosis of depressive illness in physically ill children.

Major depressive episode (MDE)	MDE is defined by the presence of five (or more) of the following symptoms occurring most of the day, virtually every day, over the course of a two-week period: depressed mood; apathy; weight change due to appetite change; insomnia or hypersomnia; indecisiveness or impaired focus; weariness or lack of energy; feeling of worthlessness or guilt; psychomotor agitation or retardation; and repeated thoughts of death or suicide.
Normal bereavement in terminally ill patients	Grief, rumination about the loss, sleeplessness, poor appetite, and weight loss may emerge. The dysphoria associated with grief is likely to fade in severity over days to weeks and it comes in waves or “pangs of grief”. Grief can be accompanied by feelings of positivity and happiness, whereas MDE is usually characterized by overwhelming sadness and misery. Grief-related thought content is often preoccupied with thoughts and recollections of the object of loss (e.g., good health), as opposed to the self-critical or gloomy ruminations observed in an MDE. In grief, self-esteem is usually maintained.
Adjustment disorder	Stressful life experiences such as chronic illness can result in psychological changes. An adjustment disorder is present when these changes are clinically significant but do not meet criteria for another mental disorder.
Depressive disorder due to another medical condition and induced by a substance or medication	A prominent and persistent period of depressed mood or anhedonia resulting from the direct pathophysiological consequence of another medical condition (e.g., hypothyroidism) or from the use of a substance/medication.
Delirium presenting as depression in a chronically ill child	Delirium may be difficult to distinguish from depression in a chronically ill child, especially when delirium has hypoactive features. In the presence of delirium, symptoms of depression have less diagnostic certainty.

Adapted from: [7,30,35,36].

**Table 4 children-09-01051-t004:** Additional considerations when selecting an antidepressant.

Hepatic	Cardiac	Renal
Generally, in patients with hepatic disease, start with a low dose and titrate slowly [43]. Furthermore, intravenous delivery of medications with substantial hepatic metabolism may bypass first-pass metabolic effects. Gastrointestinal disease as well as certain medications (e.g., those with anticholinergic activity) may affect drug absorption [44].	Congestive heart failure can interfere with medication absorption by reducing the perfusion of gastrointestinal and intramuscular drug absorption sites [44]. Orthostatic hypotension, conduction abnormalities, and arrhythmias are possible side effects of several psychotropics (e.g., tricyclic antidepressants (TCA), trazodone, etc.) Certain psychotropics (e.g., TCA, citalopram, ziprasidone, etc.) may prolong the QTc interval. Before starting therapy, patients with risk factors for sudden cardiac death should be referred for a cardiac evaluation [45].	Renal insufficiency has pharmacodynamic consequences [44]. Generally, in patients with renal failure, start a low dose with prolonged dosing intervals. In individuals with renal insufficiency, the rule of two-thirds states that drug dosages should be lowered by one-third of the regular amount. Certain psychotropic drugs may require dosage modifications in individuals with renal failure, including lithium, methylphenidate, venlafaxine, divalproex sodium, gabapentin, and topiramate.

**Table 5 children-09-01051-t005:** Non-pharmacological interventions to prevent delirium.

Placing the patient in a private room near the nursing station. Preferably having a family member or staff with the patient for supervision, interaction, and patient safety.
2. Regular and consistent reorientation and reassurance of the patient in a quiet environment.
3. Regulation of the sleep–wake cycle by optimizing lighting in the room.
4. Promoting ambulation as soon as feasible.
5. Minimizing the use of restraints.

**Table 6 children-09-01051-t006:** ABCDEF bundle.

A	Assess, prevent, and manage pain
B	Both spontaneous awakening trial and spontaneous breathing trial
C	Choice of analgesia and sedation
D	Assess, prevent, and manage delirium
E	Early mobility and exercise
F	Family engagement and empowerment

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
