# Peer review of "Psychological and Psychiatric Comorbidities in Youth with Serious Physical Illness"

_children, 2022, doi:10.3390/children9071051_

Round 1

Reviewer 1 Report

Thank you for the invitation to review this draft. It is estimated that mental disorders, including mood, anxiety and behavioural disorders, seriously affect the healthy development of children and adolescents, especially those with chronic diseases. The study analyzed common symptoms of mental disorders and provided some psychological and pharmacological advice for patient interventionswhich is of certain practical significance. However, I believe the paper can be strengthened furthermore if the study addresses some issues as follows:

The whole of the article:

1The organization of the article is a bit confusing. It is suggested that the authors integrate the definition, manifestation, treatment and pharmacological treatment of anxiety, depression and delirium.

2The common causes of anxiety, depression and Delirium are not clearly described.

3The title of the article contains other psychological distress, but it is not mentioned in the text.

Psychological Distress among Children with Serious and/or Terminal Illness:

1(Page1, Line30-33) Similarly, anxiety disorders and disruptive behavior, including defiance, oppositionality, argumentativeness, and emotion dysregulation are also common among youth with chronic illness, with prevalence rates ranging from 7 to 40%.

"from 7 to 40%" Should be replaced by "from 7 %to 40%".

2Because the article is professional, the author needs to define some proper terms (Anxiety, Depression, Delirium, children, adolescents, etc.) for readers to understand.

3(Page1, Line35-37) Specifically, parents of children with a chronic illness often have difficulty with limit setting and engage in permissive practices to offset the children’s adversities, potentially leading to increased noncompliance and “acting out” behaviors.

What is "acting out" behaviors? Can you give an example?

Psychotherapeutic Interventions for Anxiety:

1(Page3, Line91-92) Amongst different psychotherapies, CBT has the most robust evidence for childhood anxiety disorders and has been adapted to the medical setting.

Does the author have data to support this conclusion?

2(Page4, Line117-120) At a minimum, children and families are coached in recognizing “worry thoughts” when they occur and labeling them as such, in a non-judgmental manner (“That sounds like a worry thought”).

Delete the quotation marks in parentheses.

Pharmacological Intervention for Anxiety:

1In this part, the author also needs to analyze the considerations and basis for doctors and patients in choosing therapeutic drugs, rather than just describing the characteristics of drugs.

Depression:

1(Page7, Line206-208) Depressive episodes in adolescents are associated with significant morbidity and mortality, with suicide being the second leading cause of death in this age group, in the United States.

Which age group does the sentence refer to specifically, 15-19 or 10-14?

Special Considerations for Adolescents and Young Adults at End-of-Life:

1It is recommended that this section be integrated with Part 9.

Management:

1Adjust table 5 to make it more beautiful.

Author Response

Thank you for the opportunity to revise our manuscript, entitled “Anxiety, Depression, Delirium, and Other Psychological Distress in the Context of Serious or Terminal Illness.” We are very appreciative of the feedback from Reviewer 1. We have now addressed the reviewers’ comments and are eager to submit an improved manuscript for your review. Reviewer comments and detailed responses are outlined below. We appreciate your consideration of our revised manuscript and look forward to hearing from you.

Point 1: The whole of the article:

The organization of the article is a bit confusing. It is suggested that the authors integrate the definition, manifestation, treatment and pharmacological treatment of anxiety, depression and delirium.

Response 1: We have attempted, when possible, to avoid redundancy. For instance, the reader is directed to the commonality in the treatment for depression and anxiety. We have organized this manuscript by condition to allow the reader to easily link the information to individual conditions. In each section, we have defined the condition, using DSM criteria, elaborated on the clinical manifestations, and detailed evidence-based treatment interventions when available, while keeping in mind the need to remain within the word limit.

Point 2: The common causes of anxiety, depression and delirium are not clearly described.

Response 2: We agree with Reviewer 1 that a comprehensive list of the etiologies underlying anxiety, depression, and delirium is not included, in large part because of the word limit in light of the breadth of the topics covered. We call this Reviewer’s attention to several sections in the manuscript highlighting the bio-psycho-social set of factors that often contribute to the onset of mental illness and the multifactorial nature of the conditions that lead to delirium. Given the focus on physically ill children, we have kept the focus narrow.

Point 3: The title of the article contains other psychological distress, but it is not mentioned in the text.

Response 3: We have modified the title as per the suggestion of Reviewer 1.

Point 4: Psychological Distress among Children with Serious and/or Terminal Illness: 

(Page1, Line30-33) Similarly, anxiety disorders and disruptive behavior, including defiance, oppositionality, argumentativeness, and emotion dysregulation are also common among youth with chronic illness, with prevalence rates ranging from 7 to 40%. "from 7 to 40%" Should be replaced by "from 7% to 40%".

Response 4: Done.

Point 5: Because the article is professional, the author needs to define some proper terms (Anxiety, Depression, Delirium, children, adolescents, etc.) for readers to understand.

Response 5: We have included the DSM criteria that define the different disorders. We now include, in the first sentence, a more detailed age-based definition of children and adolescents.

Point 6: (Page1, Line35-37) Specifically, parents of children with a chronic illness often have difficulty with limit setting and engage in permissive practices to offset the children’s adversities, potentially leading to increased noncompliance and “acting out” behaviors. What is "acting out" behaviors? Can you give an example?

Response 6: We have addended the text to provide examples of “acting out,” including arguing, defiance, and being disruptive.

Psychotherapeutic Interventions for Anxiety:

Point 7: (Page3, Line91-92) Amongst different psychotherapies, CBT has the most robust evidence for childhood anxiety disorders and has been adapted to the medical setting. Does the author have data to support this conclusion?

Response 7: Thank you for highlighting this omission. We have now cited a review of CBT for anxiety disorders to support this conclusion (Banneyer et al., 2018).

Point 8: (Page4, Line117-120) At a minimum, children and families are coached in recognizing “worry thoughts” when they occur and labeling them as such, in a non-judgmental manner (“That sounds like a worry thought”). Delete the quotation marks in parentheses.

Response 8: Done.

Pharmacological Intervention for Anxiety:

Point 9: In this part, the author also needs to analyze the considerations and basis for doctors and patients in choosing therapeutic drugs, rather than just describing the characteristics of drugs.

Response 9: We thank Reviewer 1 for this suggestion, which has been now added.

Depression:

Point 10: (Page7, Line206-208) Depressive episodes in adolescents are associated with significant morbidity and mortality, with suicide being the second leading cause of death in this age group, in the United States. Which age group does the sentence refer to specifically, 15-19 or 10-14?

Response 10: We apologize for the confusion and have clarified in the text that it is referring to 10- to 19-year-olds. We have also edited the statement by adding “children” to it.

Special Considerations for Adolescents and Young Adults at End-of-Life:

Point 11: It is recommended that this section be integrated with Part 9.

Response 11: We have now integrated this within the earlier section.

Management:

Point 12: Adjust table 5 to make it more beautiful.

Response 12: We have left aligned the content of Tables 5 and 6.

Reviewer 2 Report

Thank you for inviting me to review this manuscript, titled “Children Special Issue on Pediatric Palliative Care: Anxiety, Depression, Delirium, and Other Psychological Distress in the Context of Serious or Terminal Illness”.  This study propose to provide a review of the symptoms that comprise mental conditions often observed in children with chronic illness or at the end of life, and some guidance to help clinicians distinguish normative from pathological presentations.

The specific issue or problem is defined.

The main question addressed by the research is relevant and interesting.

The proposed objective is relevant, as well as the evidence provided and the accompanying discourse.

However, the manuscript would benefit from some small suggestions or changes. See specific suggestions below.

I would try to shorten the length of the title

It would be necessary to alphabetize the keywords in the abstract unless you want to emphasize some of them

It would indicate more clearly the type of study involved, the design and the methodology

If it is a review work, for example, it would be necessary to indicate the databases in which the search has been carried out, the flow chart, inclusion and exclusion criteria of the articles selected for the information query, etc.

It is recommended to follow the classic structure of introduction, method, results, discussion, conclusions, limitations, future lines. For example, a section on conclusions, future lines of research and limitations is not included.

Expand the limitations of the study and some other final paragraph with the implications of the present study

Author Response

Thank you for the opportunity to revise our manuscript, entitled “Anxiety, Depression, Delirium, and Other Psychological Distress in the Context of Serious or Terminal Illness.” We are very appreciative of the feedback from Reviewers 2. We have now addressed the reviewers’ comments and are eager to submit an improved manuscript for your review. Reviewer comments and detailed responses are outlined below. We appreciate your consideration of our revised manuscript and look forward to hearing from you.

Point 1: I would try to shorten the length of the title.

Response 1: We have modified the title to “Psychological and Psychiatric Comorbidities in Youth with Serious Physical Illness.”

Point 2: It would be necessary to alphabetize the keywords in the abstract unless you want to emphasize some of them.

Response 2: We have now alphabetized the keywords.

Point 3: I would indicate more clearly the type of study involved, the design and the methodology. If it is a review work, for example, it would be necessary to indicate the databases in which the search has been carried out, the flow chart, inclusion and exclusion criteria of the articles selected for the information query, etc.

Response 3: This invited manuscript is a narrative review, as opposed to a systematic review, which is the reason the requested details were not included.

Point 4: It is recommended to follow the classic structure of introduction, method, results, discussion, conclusions, limitations, future lines. For example, a section on conclusions, future lines of research and limitations is not included.

Response 4: We agree with Reviewer 1 that the proposed structure would have been necessary for a systematic review. However, given that his manuscript is a narrative review, the content does not lend itself to the classic structure and would be more easily understood by the reader if organized by topic instead. This is in keeping with other chapters similarly intended (e.g., Schuelke et al., 2021).

Point 5: Expand the limitations of the study and some other final paragraph with the implications of the present study.

Response 5: We thank the Reviewer for this comment. Because this is a narrative review, including a “Limitations” section does not apply. However, we have expanded the final paragraph to comment on the need for more research that specifically includes children with comorbid medical and psychiatric conditions.

Round 2

Reviewer 1 Report

The author has made modifications according to my suggestions, so I suggest that this paper be accepted.

Reviewer 2 Report

The t of Table 2 within the text must be capitalized